# TEfinder: A Bioinformatics Pipeline for Detecting New Transposable Element Insertion Events in Next-Generation Sequencing Data

**DOI:** 10.3390/genes12020224

**Published:** 2021-02-04

**Authors:** Vista Sohrab, Cristina López-Díaz, Antonio Di Pietro, Li-Jun Ma, Dilay Hazal Ayhan

**Affiliations:** 1Department of Biochemistry and Molecular Biology, University of Massachusetts Amherst, Amherst, MA 01003, USA; vsohrab@umass.edu (V.S.); lijun@biochem.umass.edu (L.-J.M.); 2Departamento de Genética, Universidad de Córdoba, 14071 Córdoba, Spain; g02lodic@uco.es (C.L.-D.); ge2dipia@uco.es (A.D.P.); 3Molecular and Cellular Biology Graduate Program, University of Massachusetts Amherst, Amherst, MA 01003, USA

**Keywords:** transposable elements, mobile element insertion events, next-generation sequencing (NGS), genome evolution

## Abstract

Transposable elements (TEs) are mobile elements capable of introducing genetic changes rapidly. Their importance has been documented in many biological processes, such as introducing genetic instability, altering patterns of gene expression, and accelerating genome evolution. Increasing appreciation of TEs has resulted in a growing number of bioinformatics software to identify insertion events. However, the application of existing tools is limited by either narrow-focused design of the package, too many dependencies on other tools, or prior knowledge required as input files that may not be readily available to all users. Here, we reported a simple pipeline, TEfinder, developed for the detection of new TE insertions with minimal software and input file dependencies. The external software requirements are BEDTools, SAMtools, and Picard. Necessary input files include the reference genome sequence in FASTA format, an alignment file from paired-end reads, existing TEs in GTF format, and a text file of TE names. We tested TEfinder among several evolving populations of *Fusarium oxysporum* generated through a short-term adaptation study. Our results demonstrate that this easy-to-use tool can effectively detect new TE insertion events, making it accessible and practical for TE analysis.

## 1. Introduction

Transposable elements (TEs) are DNA sequences that move from one genomic location to another and thus impact genome evolution and organism adaptation [1]. TE transposition can alter the genomic architecture, introduce structural polymorphisms, disrupt coding sequences, and affect transcriptional and translational regulation. Additionally, TEs are capable of changing eukaryotic gene expression by providing cis-regulatory elements such as promoters, transcription factor binding sites, and repressive elements [2,3]. Ultimately, TEs provide a wide array of genomic diversity, functional impact, and evolutionary consequences that can be of notable interest to population genetics, host interaction, and comparative genomics studies.

TEs comprise a significant portion of the genome of humans and many other organisms, due to their mobilization and accumulation throughout evolution [4]. While some TEs are no longer active, certain TE families remain mobile and their transposition contributes to genetic variation both at the individual and the population level. TEs play important roles in many biological processes, such as cancer biology [5], neurodegenerative diseases [6], or host-pathogen interactions [7]. Therefore, it is of high interest to identify transposon insertion polymorphisms (TIPs) for the detection of highly active TE families and the understanding of their contribution to genome dynamics and organism adaptation.

Advancements in next-generation sequencing technologies have made in silico discovery of transposon insertion events readily accessible. Two features commonly used for the detection of TE insertions are the target site duplication (TSD) and discordantly aligned reads. Upon insertion of certain transposons into a new genomic location, the mechanism of integration results in the duplication of the target sequence at the integration site, which is referred to as the TSD [4]. TSD length varies across superfamilies, and the identification of this structural motif proves useful in determining true transposition events within the genome [8]. Due to the nature of the transposon insertion, the TE sequence should be mapped to existing TE locations within the genome, while the paired-end read partner should be mapped to the unique sequence at the insertion site. This will result in discordant reads, which can be recognized when two paired-end reads are placed in different genomic locations or in a much greater distance that exceeds the expected insert size of the sequencing library. The more discordant reads localized in a genomic region (cluster), the higher the confidence to call it a new insertion site.

Although several bioinformatic tools that detect such events have been developed, the broad application of these tools is limited by heavy external software or file dependencies. For instance, ISMapper [9] can report insertion positions of bacterial insertion sequences (ISs) when provided with paired-end short-read sequences, TE sequences as multi-FASTA queries, and the reference genome sequence. However, this requires specific versions of Python 3 and BioPython, which may not be readily available to all users and lead to difficulties in running the software. Mobile Element Locator Tool (MELT) [10], a Java software package originally developed as a part of the 1000 Human Genomes Project [11], discovers, annotates, and genotypes mobile element insertions with the only requirement of Bowtie2 [12]. The package is powerful in comprehensive TE analysis in human and chimpanzee genomes. However, it requires gene annotations. For each TE, users need to provide the consensus sequence and multiple files related to this particular transposon. Users have to make sure only “ATGC” characters are present in the consensus FASTA file. Such a high level of external file dependency makes it challenging for users who are interested in TE analysis in non-model organisms that lack well-annotated genomes.

Encountering difficulties in applying available TE insertion detection tools, we developed a simple bash bioinformatics pipeline, TEfinder, to detect new insertion events using tools that are commonly embedded in genomics variant calling workflows, including BEDTools [13], SAMtools [14], and Picard [15]. Required input files include the reference genome sequence, TE annotations of the reference genome sequence, alignment of paired-end short-read sequencing data, and a list of TE names of interest. The pipeline reports new insertion events based on the TE annotation of the reference genome sequence. The output file can be in either BED or GTF format that captures all details and can be integrated into the downstream analysis. Here we report the design of the pipeline, testing results of its performance using short-read sequencing data derived from a short-term evolution experiment in the filamentous fungus *Fusarium oxysporum* f. sp. *lycopersici* 4287 (Fol4287), as well as a simulation dataset from chromosome 2 L of *Drosophila melanogaster* [16], and two datasets from the *Arabidopsis thaliana* mobilome study [17,18].

## 2. Materials and Methods

### 2.1. Requirements

TEfinder is a bash pipeline for detecting TE insertions using paired-end sequencing data. The overall objective is to identify new TE insertion events in a given sample that are different from ones captured in the reference genome sequence. For this, an assembled genome and pair-end sequencing of the sample is required.

Software required to run this tool include BEDTools 2.28.0 or later [13], SAMtools 1.3 or later [14], and Picard 2.0.1 or later [15]. The four user input files are a FASTA file of the reference genome sequence, a file of paired-end read alignments (BAM or SAM), TEs present in reference genome sequence in GTF format, and a list of TE names to be analyzed in text file format. The reference genome sequence and the read alignment files are essential. The TE GTF file can be produced using any annotation tool (see Implementation for an example). The list of TE names provides users the option to focus their analysis on selected TE families. Without a specification, the list can be easily derived from the TE GTF file.

Optional arguments have been incorporated for users to customize the tool.

The default DNA fragment length or insert size of the short-read sequencing library is 400 base pairs (bp). For a sequencing library with an insert size of 500, this value can be modified by including “-fis 500”;The default maximum distance between reads for merging and forming clusters has been set to 150 bp. To decrease the value of this parameter by 30 bp, the “-md” argument can be set to 120;The default maximum target site duplication (TSD) length is 20 bp. Modifying this value can be useful if the TSD lengths of the TEs being analyzed are known leading to more targeted TE analysis results. The maximum TSD length can be increased to 30 bp by setting “-k 30”;The “-picard” argument should be set to the full path of the picard.jar file.An additional Java argument relating to Picard’s maximum memory heap size can be submitted to the pipeline as a fraction of the total memory allocated, leading to enhancement of the overall runtime. A maximum memory heap size of 25,000 MB is set via “-maxHeapMem 25000”;Multithreading is supported for the SAMtools commands of the pipeline via the thread option. The number of threads can be set to 4 using “-threads 4”;A working directory name can be provided as such “-workingdir TEfinder_Y1” for better file organization and differentiation amongst TEfinder runs;An output name can be provided to be appended to the default output names for effective labeling of files, such as “-outname Y1”;Users can also specify GTF as the output argument, “-out gtf” (case insensitive), which reports the TE insertions in GTF format;Lastly, if the user includes the optional argument “-intermed yes” (case insensitive), all intermediate files including the TE-specific directories are provided in addition to the output files.

### 2.2. Implementation

#### 2.2.1. Preprocessing

Preparation of the four input files depicted in the top panel of Figure 1A:A FASTA file of the reference genome sequence. No special requirement.A BAM file of aligned paired-end reads to the reference genome sequence.

The sample sequencing reads needed to be aligned to the reference genome sequence using an aligner. In this study, Burrows–Wheeler Aligner (BWA) [19] was used. Users may choose other aligners, as long as read group information is included in the header created by the aligner. Different aligners can have slightly different results.


    bwa index reference.fasta
    bwa mem -R “@RG\tID:sample\tPL:illumina\tLB:LIB\tSM:sample” \
    reference.fasta sample_R1.fq sample_R2.fq > sample.bam


The bwa command above specifies read group information using bwa-mem, option “-R”. If Bowtie2 [12] was used, and users needed to select option “--rg-id”. For GEM3-Mapper [20], option “-r” needed to be specified. If read group information was not included in the initial alignments, Picard [15] has AddOrReplaceReadGroups function to create read group information based on the input BAM file.

3.GTF file of TE annotation in the reference genome sequence.

This file captured genomic locations of all TEs that were present in the reference genome sequence. This file could be GFF2/GTF or GFF3. To generate it, a library of TE sequences of the reference genome had to be provided. For a well-annotated genome, users could obtain this library from existing databases of known repetitive elements such as Repbase [21]. If such a library is not available, users can compile one by running a de novo TE family identification software such as RepeatModeler [22], RepeatScout [23], or RepeatMasker [24] on the reference genome sequence. An example is shown below to use RepeatScout to discover repetitive sequences and form the TE library of the reference genome sequence.


    build_lmer_table -sequence reference.fasta -freq reference.freq
    RepeatScout -sequence reference.fasta -output TElib.fa -freq \
    reference.freq


The most straightforward tool was RepeatMasker [24], which is available on Galaxy [25], and could be used to generate the GTF file based on the provided reference genome sequence and a common repeat library from RepBase [21] or the TE library created above. The output from RepeatMasker was to be filtered so that simple repeats would be removed.


    RepeatMasker -lib TElib.fa -dir workingdir -gff reference.fasta


4.Text file with names of the TEs of interest.

The TE names provided needed to exactly match the names in the GTF entries in a single column. The names could not include characters other than letters, digits, “-”, “_”, and “#”. More information about this can be found in the tool manual.

#### 2.2.2. Pipeline

TEfinder relies on paired-end sequencing and uses information on discordant reads, which are reads that do not match the expected orientation or insert size. The software required for the package are SAMtools [14], BEDTools [13], and Picard [15].

A typical command to run TEfinder is:


    TEfinder -alignment sample.bam -fa reference.fa -gtf TEs.gtf \
    -te List_of_TEs.txt


The reference sequence information anchors all downstream analysis in TIPs, and the initial step of the tool was creating the FASTA index file which enhances efficient access to regions within the FASTA file. The subsequent step of the tool was to sort the input alignment file by coordinates. To avoid multi-mapping and chimeric ambiguities from affecting our analysis, the tool removed secondary and supplementary alignments from the user-provided sample alignment file (Figure 1A). For each TE, the package went through the following logical steps:Identify discordant reads. The program extracted all primary reads mapped to known TEs in the reference using BEDTools intersect. Then, alignments of the selected reads and their pairs were extracted from the input BAM file using the FilterSamReads tool of Picard [15]. Among those, reads were selected as discordant if the corresponding mate mapped to a different sequence or the read pair had an insert size that exceeded a threshold of 10 times the mean insert size (Figure 1B, top panel).Group discordant reads. Once the discordant read alignments had been filtered, the regions of clustered reads were identified using the BEDTools merge so that reads aligned to the plus strand and minus strand were grouped separately (Figure 1B, middle and bottom panels). In this step, the reads had to be overlapping or within a given distance to be considered in the same group.Define a TE insertion site. Each plus-strand group was coupled with the nearest minus-strand group. The coupled regions went through a filtering step to remove incorrect orientations, considering that TE sequences should be present between forward and reverse groups and not the reverse order. Due to the nature of duplication of the target site upon the transposon insertion, TSD sequences may have been present in both forward and reverse strands, resulting in an overlap between forward and reverse clusters. If an overlapping site was smaller than the maximum TSD length, the location was reported as a possible TE insertion site. If they were not overlapping due to low coverage or other reasons, and the distance between the groups was smaller than the given threshold, the region in between was reported.Filter new insertion sites. The identified insertion sites were filtered based upon 3 criteria: Occurrence in repeat regions, supporting read count, and strand bias. If the insertion site coincided with a TE-annotated site in the reference genome sequence, its filter label became “in_repeat”. If the total supporting reads for the insertion event was less than the cutoff 10, then the filtering process labeled this low confidence event as “weak_evidence”. Strand bias was critical for filtering out insertion events. Two threshold values were calculated based on the forward read count (F), and if the reverse read count (R) fell outside those boundaries (R < F^0.8^ or R > F^1.25^), then the event likely was not caused by TIP and received a “stand_bias” label. If an insertion site passed all three filtering criteria, then it was labeled “PASS”, representing high confidence insertion events (Figure 1).

Detected TE insertions events were reported in BED detail format with 7 columns: (1) Insertion site sequence, (2) start coordinate, (3) end coordinate, (4) TE family, (5) total number of reads supporting the insertion event, (6) unused, (7) additional information regarding the insertion such as the number of forward and reverse reads supporting the event (FR and RR), as well as the insertion region start and end coordinate (InsRegion), and the filtering results (FILTER) obtained from the pipeline. Additionally, discordant pair alignment files from individual TE families were combined to create the “DiscordantReads.bam” file in the working directory (Figure 1B, middle panel). This file could be used to visualize the events on genome browsers such as Integrative Genomics Viewer (IGV) [26]. If the GTF output option was selected, the GTF file reported the additional information as attributed in the ninth column.

### 2.3. Testing Dataset and Processing

The performance of TEfinder was tested using a model *Fusarium oxysporum,* a soil-inhabiting ascomycete fungus that causes devastating losses in more than a hundred different crops and disseminated infections in immunocompromised humans [27,28]. One interesting genomic feature of the *F. oxysporum* species complex is the compartmentalization of its genome, where conserved core regions carry essential house-keeping functions while lineage-specific accessory regions are enriched for TEs and associated with host-specific pathogenicity [28,29].

The pipeline was used to identify new TE insertion events among five populations, Y1–Y5, evolved under laboratory conditions. Briefly, the ancestor strain (WT) of Fol4287, which was previously used to generate the reference genome assembly [30], was subjected to successive transfers on yeast peptone dextrose agar plates. After 10 passages with 5 independent biological replicates (Y1–Y5), genomic DNA was extracted from the final mixed populations and sequenced using Illumina HiSeq 2500 platform with 2 × 71 cycles. The whole-genome shotgun sequencing reads are available at NCBI under project PRJNA682786 and datasets SRR13203443, SRR13203444, SRR13203445, SRR13203446, and SRR13203447.

To assess the sensitivity, we performed random sampling of the alignments in the Y2 population using Picard [15] and reduced the coverage to 50%, 30%, 20%, and 10% of the initial read coverage and repeated the analysis.

A small subset of simulation data from chromosome 2 L of the *Drosophila melanogaster* reference genome sequence (dm3), used for testing TEMP [16] software available on GitHub at (https://github.com/JialiUMassWengLab/TEMP, accessed on 28 January 2021), was also used to evaluate TEfinder.

To test TEfinder on identifying new insertion events in 2 *Arabidopsis thaliana* accessions [17], Alst-1 (SRR492202) and Benk-1 (SRR492214) whole-genome paired-end sequencing data were mapped to the TAIR10 reference genome (ftp://ftp.arabidopsis.org/home/tair/Genes/TAIR10_genome_release) with BWA [19]. The median sequencing coverage for Alst-1 and Benk-1 were 36 and 40, respectively. TAIR10 TE annotation GTF was obtained using RepeatMasker [24] and the TAIR10 transposon library available at (ftp://ftp.arabidopsis.org/home/tair/Genes/TAIR10_genome_release/TAIR10_transposable_elements/, accessed on 28 January 2021). Non-reference insertion events for these accessions, reported in an *A. thaliana* mobilome analysis [18], were used for benchmarking.

### 2.4. Experimental Validation

Thirteen reported TE insertion events were validated by PCR. Genomic DNA was extracted from mycelia of the *F. oxysporum* reference strain, evolved populations, or of single spore (SS) isolates obtained from the experimentally evolved lines, using the Cetyltrimethylammonium Bromide method [31]. PCR was performed in a thermocycler using the thermostable DNA polymerase of the Expand High Fidelity PCR System (Roche Diagnostics, Mannheim, Germany). Each PCR reaction contained 300 nM of each primer, 2.5 mM MgCl_2_, 0.8 mM dNTP mix, 0.05 U/μL polymerase, and 5–10 ng/μL genomic DNA. PCR cycling conditions were as follows: An initial step of denaturation (5 min, 94 °C); 35 cycles of 35 s at 94 °C, 35 s at the calculated primer annealing temperature, and 1 min/1.5 kb extension at 72 °C (or 68 °C for templates larger than 3 kb); and a final extension step of 10 min at 72 °C (or 68 °C). For each predicted TE insertion event, a pair of specific primers flanking the insertion site was designed.

## 3. Results

### 3.1. Data Preparation

The Fol4287 reference genome assembly was 53.9 MB in 499 scaffolds [30]. The sequencing reads were mapped to the reference using BWA [19] with >99% mapping and median coverages ranging from 67 to 94 (Table 1). RepeatMasker [24] was used to identify the known TEs in the genome with a curated TE library [27,28] which included 69 TE families. Approximately 4.5% of the entire genome was identified as repetitive sequence, with 3.98% of the genome comprised of transposable elements. Accessory sequences included 74% of all TEs in the genome [27].

### 3.2. Total TE Insertion Events Detected

On a high-performance computing cluster with a memory request of 3 cores × 50,000 MB, the trial runs took an average of 4.7 h to complete with the minimum time being 3.6 h across the samples. Three threads were provided for SAMtools multithreading. The maximum heap memory for Java was set to 25,000 MB to enhance the run time when filtering the alignment to only include discordant reads.

TEfinder detected 502 to 570 insertion events across whole genome sequencing data from 5 evolved Y1–Y5 populations. Details for each individual sample are summarized in Table 2 (Appendix A). The TIPs had varying allele frequencies depending on the population structure. As expected, many insertion events were detected among complex repetitive regions that captured most transposons. In fact, the number of new TE insertions reported in each sample population was approximately one-third of the insertions detected in known repeat regions. Additionally, there were filtered out events because of the low read count (“weak evidence”). Furthermore, almost half of the detected events had strand bias as estimated by a power function. The constants for this function were determined arbitrarily for the datasets tested in this study. The users could utilize the reported forward and reverse read counts for their data if the need arose.

The BAM output file format was intended for visual inspection. Figure 1B captures a new insertion event of the TE small Hornet (TIR/hAT) that reached fixation in the Y3 population visualized in IGV. The confidence level for calling this new insertion event was high, with a total of 415 supporting reads spanning a 1114-bp insertion region in a dataset with 92× median coverage. Of these supporting discordant reads, 223 reads were grouped in the plus strand cluster and 192 in the minus strand cluster. Since the TEfinder pipeline does not focus on TSD position detection, the reported insertions may not always coincide precisely with the true location. Nevertheless, the pipeline mapped the TSD position of many events precisely. In the example of Figure 1B, two clusters overlap at an 8-base TSD, which coincides with the known TSD of this particular transposon superfamily [8].

We selected 13 newly detected TIPs in the “PASS” catalog for experimental validation (Table 2). Results of PCR confirmed our prediction in 12 out of 13 cases, resulting in a success rate of 92%. The one TIP event that failed PCR amplification only had 22% allele frequency in that evolved population. The success rate for non-PASS events might be lower. Figure 2 shows four TIPs by small Hornet in Y1, Y3, Y4, and Y5 populations. The size difference in the ancestor and the lines with TIP were about 750 bp, which coincided with the size of the small Hornet transposon.

### 3.3. Sensitivity and Applicability of TEfinder

The output BED file of TEfinder reported an overview of the total number of reads supporting the insertion event, as well as the number of forward and reverse read counts. In our trial, TEfinder was able to capture low-frequency insertion events with read evidence as little as 8 in all datasets.

To test the sensitivity of TEfinder, we used one of the evolving samples with read coverage reduced to 0.5, 0.3, 0.2, and 0.1 of the original dataset (Table 3, Figure 3, Appendix A), corresponding to 47×, 28×, 19×, and 9× sequence coverage. When the read coverage reduced to 47×, 82.5% of the previously detected events were captured. At 9× coverage, only 38% of the events were detected. Therefore, the sensitivity of TEfinder depended on the read coverage of the alignments in the input BAM file. To ensure effectiveness in the detection of TIPs, we suggest a minimal 20× sequence coverage, especially when reads are generated from a mixed population.

We used TEfinder with default options except “-k 30” on the subset of simulated data from chromosome 2 L in *D. melanogaster* [16]. There were 10 simulated insertions and 8 were by TEs present in chromosome 2 L. TEfinder identified all 8 new insertions. Seven of the eight events were in the “PASS” category and one event was labeled as strand biased with 32 forward and 14 reverse read counts (Appendix A). Since the other two simulated insertion events were caused by TEs from other parts of the genome and absent in the chromosome 2 L TE annotation, they were not detected by the TEfinder pipeline. Furthermore, we tested TEfinder with data used in the *Arabidopsis thaliana* mobilome study [17,18] that detected 63 non-reference TE insertion events in Alst-1 and 61 in Benk-1 accessions. With default options, TEfinder successfully detected 55 and 58 of these known instances, respectively. These testing results demonstrated feasibility in applying TEfinder in plants, animals, and fungi with high sensitivity.

## 4. Discussion

Here we reported a simple pipeline to detect new TE insertions in related populations when compared to a reference genome sequence. The required input and output file types are commonly used in variant detection workflows and therefore it can be easily implemented in larger pipelines. The pipeline has a low RAM requirement and takes a rather short computation time. Testing of the pipeline in evolved populations of the fungus *F. oxysporum* demonstrated that this easy-to-use tool could effectively detect new TE insertion events, making it accessible and practical to address TE activity-related biological questions in population genomics, genome evolution, and other applications. Successfully detecting TIPs using *Drosophila melanogaster* and *Arabidopsis thaliana* data demonstrated feasibility in applying TEfinder in plants, animals, and fungi with high sensitivity.

Paired-end sequence reads are the foundation to read out genetic changes in an individual or a population when comparing to a reference genome sequence. Therefore, sufficient sequence coverage and good quality of the sequence reads are important, as for all variant calling software. However, even with a small number of read evidence, the pipeline can still capture low-frequency events. The optional parameters can be used to adjust the strictness of the algorithm.

Our data suggest that as long as the reference genome sequence is reasonably assembled, the quality of the reference genome assembly should not affect the performance of the TEfinder pipeline. For instance, the assemblies of the lineage-specific genome regions of the Fol4287 genome used to test TEfinder were fragmented due to high repeat content, while chromosome-level assembly was accomplished for most core regions. Importantly, TEfinder was able to capture the events in both types of regions. TEfinder depends on read placement within a genome. Ambiguity exists in such processes especially when placing a read into a highly repetitive region, which may result in lower confidence level in calling a TIP. Such insertion events, as well as nested TEs, might still be detected and reported by TEfinder with a tag “in repeat”, to differentiate them from TIP events with high confidence.

As with other variant calling software, the output files need to be filtered before further analysis. Users can utilize the internal filter tags and reported forward, reverse, and total read count values according to their needs. The output BAM file is also useful to do visual confirmations. One feature missing from the output is the allele frequency of the events. However, the read counts and insertion region positions can be used to estimate allele frequencies. We were able to experimentally verify some of the events. Although we are in the early phase of understanding the functional impact of these TIPs, the ability to detect these events with high confidence enables hypothesis generation and establishment of targeted functional studies. Testing of the TEfinder pipeline in evolved fungal populations further confirmed the effectiveness of this tool in identifying TE insertion events in non-reference eukaryotic genomes.

## 5. Conclusions

TEfinder is a tool for detecting new TE insertions in fungal, plant, or animal genomes via paired-end resequencing data. TEfinder has a small number of external software requirements and input files, making it an easy-to-use and accessible tool for the detection of new TE insertions events.

## Figures and Tables

**Figure 1 genes-12-00224-f001:**
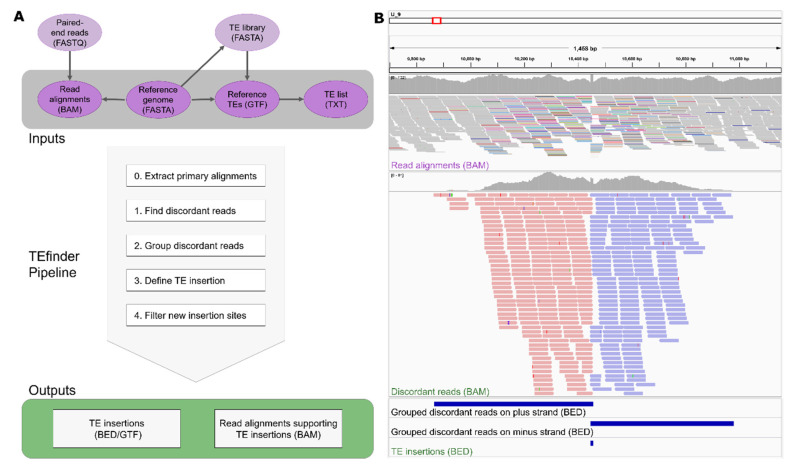
(**A**) Basic workflow of the TEfinder pipeline. The required input files are in dark purple, while other files that are used to create the input files are in light purple. Arrows show the dependencies of the files. Please check Implementation for details. The steps of the pipeline are in the light gray box and the outputs are in the green box. (**B**) Visualization of a new small Hornet (TIR/hAT) insertion event using Integrative Genomics Viewer (IGV). The data were derived from the Y3 population after a short-term evolution experiment using Fol4287. (**Top**) input read alignments file, (**middle**) output discordant reads alignment file, and (**bottom**) intermediate and output BED files are displayed. The aligned reads in the top panel are shown in squished mode with reads mapped to different sequences colored non-gray. The aligned reads in the middle panel are shown in collapsed mode and forward reads are in red while reverse reads are in blue. The histograms above the alignments in the top and middle panels show the aligned read counts. The position of the insertion event reported by TEfinder coincides with the duplicated target site (TSD).

**Figure 2 genes-12-00224-f002:**
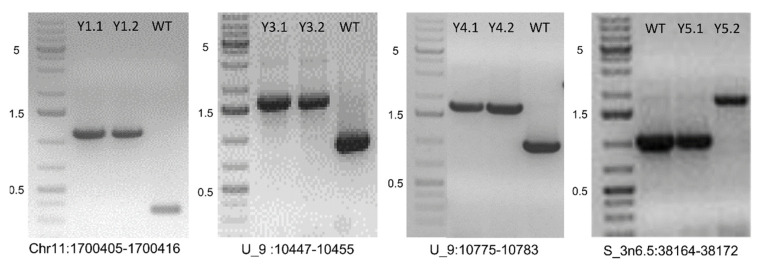
Validation of small Hornet transposon insertion events detected in experimentally evolved populations of Fol4287. PCR was performed with primers flanking the insertion site in the ancestor strain (WT) and two single spore (SS) isolates in populations Y1, Y3, Y4, and Y5, respectively. GeneRuler 1 kb Plus DNA Ladder shown with indicated sizes in kb to the left. Specific primers designed for validation of TE insertion events are Y1 forward: TCCTCCTGGGTTTCTTGTCAC, Y1 reverse: CTCTTGAAACGTGGTGCAGAC; Y3 and Y4 forward: AGACGGACAAAGAGGTGTGAC, Y3 and Y4 reverse CTCGACACTACCAGGCACTAT; Y5 forward: GAGTATGCTTCCCGATCCTTG, Y5 reverse: GACATCCCTCAATCCGCTGAA.

**Figure 3 genes-12-00224-f003:**
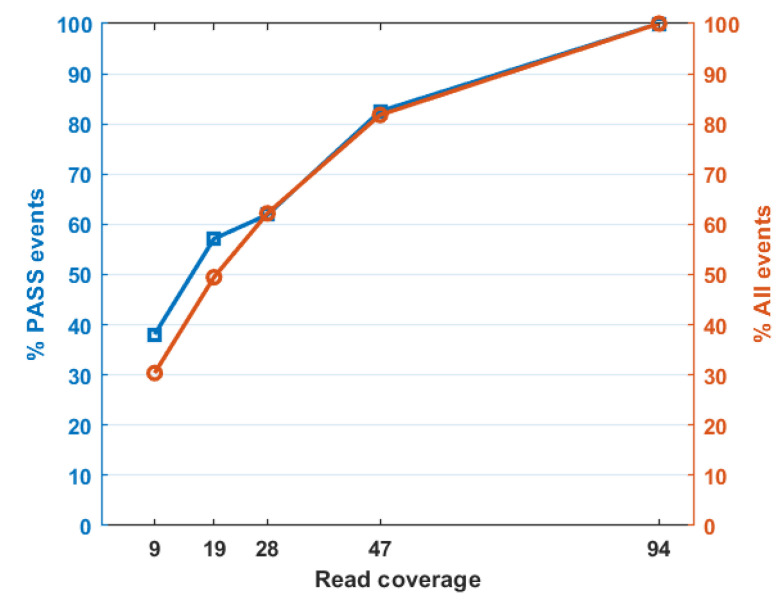
Percent of detected TIP events with respect to read coverage in the input alignment file. Orange circles depict all reported events while blue triangles represent PASS events.

**Table 1 genes-12-00224-t001:** Sample sequencing and mapping summary statistics.

Sample	Total Reads	Discordant Mate Mapping Quality ≥ 5	Percent Reads Mapping to Reference	Median Coverage
Y1	62,015,365	475,966	99.41	67
Y2	85,688,005	600,267	99.38	94
Y3	83,109,424	578,959	99.44	92
Y4	70,322,907	475,591	99.42	78
Y5	76,034,020	567,823	99.37	83

**Table 2 genes-12-00224-t002:** Number of transposable element (TE) insertion events reported by TEfinder in evolved populations of Fol4287.

Sample	All Events	In Repeat	Weak Evidence	Strand Bias	PASS	PCR-Verified
Y1	502	397	11	256	55	1/1
Y2	566	449	9	281	63	1/1
Y3	570	455	8	264	60	1/1
Y4	565	446	10	278	60	3/3
Y5	544	423	10	272	64	6/7

**Table 3 genes-12-00224-t003:** Number of TE insertion events reported by TEfinder in Y2 population with reduced coverages.

Median Read Coverage	Reported	PASS
94	566	63
47	463	52
28	352	39
19	280	36
9	172	24

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
