# Peer review of "TEfinder: A Bioinformatics Pipeline for Detecting New Transposable Element Insertion Events in Next-Generation Sequencing Data"

_genes, 2021, doi:10.3390/genes12020224_

Round 1

Reviewer 1 Report

Sohrab et al. describe a shell script (TEfinder) for the detection of novel TE insertions based on a reference genome sequence and a paired-end read mapping. There are already several tools for the detection of TE insertions, but the novel objective of TEfinder is useability. I believe TEfinder could be a helpful solution for biologists who have only basic bioinformatics skills. However, there are still some issues and further demonstration of the capacities of TEfinder is needed.

1) The analyzed fungal genome is small. The presented use case is also restricted to just one taxon. What is about larger data sets and different taxa e.g. plants and animals? I would like to see that TEfinder is also able to handle larger and different datasets. Therefore, I suggest to analyze these model systems in addition:

a) The most important plant model species is Arabidopsis thaliana with Columbia-0 as the reference accession. Accession specific differences have been characterized e.g. for Niederzenz-1 (Nd-1) and there are even some PCR validations of such differences. It would be important to show that TEfinder is able to recover these. A TE annotation is available for Columbia-0 and should be a good test if TEfinder is able to handle any GFF file.

https://doi.org/10.1371/journal.pone.0164321

https://doi.org/10.1371/journal.pone.0216233

b) The ultimate test is the analysis of an animal genome. I would suggest to take one of the latest phased long read assemblies and some randomly selected samples:

https://doi.org/10.1038/s41587-020-0719-5

https://doi.org/10.1073/pnas.1613365113

Alternatively the analysis of a Drosophila dataset could also show the applicability to animals:

https://doi.org/10.1093/nar/gkaa450

https://dx.doi.org/10.12688%2Ff1000research.9912.3

2) When analyzing species where a long read genome assembly is available for multiple accessions, it is possible to perform a benchmarking through whole genome alignment. The authors might want to consider this as an alternative approach.

3) Specific software versions are not necessarily an issue, but BioPython might not be able to process very large datasets if everything is loaded into memory. The authors might want to check if this would be another motivation for developing a new tool.

4) I do not understand the criticism about MELT. A FASTA and a BED file are common file formats and TEfinder also requires some kind of prior knowledge about TEs.

5) The authors state that the main objective of TEfinder is useability. While this is a good justification for the development of a new tool, I still see some issues. BEDTools, SAMtools and Picard are already more requirements than e.g. Bowtie2.

a) Is it possible to reduce the requirements to just samtools which is available as a jar file and does not require installation?

b) The FASTA index file should be created by TEfinder to reduce the number of required input files.

c) There should be instructions/suggestions for a very simple TE annotation approach. I assume this could be a major obstacle for the analysis of a novel genome assembly. Installing and running RepeatModeler or RepeatScout is probably not an option, because their use is challenging. Repbase is not freely accessible and a license is expensive. If users are able to deal with these tools, they could also use MELT for the following analysis. There needs to be a very convenient solutions which comes with an executable binary and does not require any installations.

d) TEfinder should support BAM and SAM files to make it easier to use. samtools could be used for internal conversion of SAM to BAM if necessary.

e) Mate pair reads (not just PE) should be supported. It is possible to convert them into paired-end. This increases the number of datasets that can be analyzed. Mate pair reads might even be beneficial when it comes to repetitive regions.

f) TE names are probably a critical point, because spelling mistakes could cause TEfinder to break. Are there any forbidden characters which would break TEfinder like space, tab, comma, pipe, etc.? The authors might want to provide some additional scripts to generate this TE name list from the annotation file or do this internally to avoid another input. If a list is not specified, all TEs should be analyzed automatically.

6) The authors state "TEs in GFF", but there are probably some more specific requirements. Was TEfinder tested with a diverse set of different GFF files? What is about GTF support? There are different versions of the GFF standard and their structure is surprisingly diverse.

7) Is it possible to use multiple threads in TEfinder e.g. for samtools?

8) The input is a GFF file, but the output is BED. It would be good to offer GFF as an alternative/additional output. Users might want to run down-stream analyses which require a GFF and not a BED file. The targeted users are probably not able to convert BED to GFF.

9) There is a difference between the "reference genome" (DNA molecules) and the "reference genome sequence" (strings of characters, assembly). Since TEs are often not completely resolved in assemblies, it is crucial to differentiate between both. I noticed this issue throughout the manuscript. Please check the manuscript carefully and correct this.

10) TEfinder uses paired-end information, but long reads (ONT, PacBio) should be even better. It would be possible to break long reads into paired-end fragments and feed them to TEfinder. The distances between both fragments would be precisely known. However, it might be better to have additional support for long reads. The authors might want to consider this, because most (re-)sequencing projects will use long reads in the future.

11) Does the applied alignment tool influence the results? Some of the common aligners like Bowtie2, BWA-MEM, Novoalign, GEM3, etc. should be evaluated to ensure that TEfinder is working as expected. There might be differences concerning the alignment score, inclusion/exclusion of unpaired reads, and the handling of multi-mapped reads.

12) What is about reads mapping in opposite directions or on different contigs? The pipeline description does not mention these cases.

13) Is there a minimal coverage that is required for this analysis? Is it possible to set a minimal coverage as filter? It would be good to evaluate subsets of reads to identify a minimal required coverage for accurate results.

14) Are nested TEs an issue for TEfinder? The two groups of reads might not be clear if there are multiple TE copies inserted nested into an existing TE array. I think this is partly addressed by the two output files, but the authors might want to provide some additional details.

15) The authors might want to add some additional details about the core concept of identifying and filtering novel TE insertions. Are there any cutoff values involved?

16) The run time looks reasonable. The details about memory usage are fine, but how many cores were used? Is it possible to run analyses in parallel e.g. run analysis for different contigs in parallel?

17) How many of the inspected insertions were correctly predicted by TEfinder? This would reveal the true positive rate. What is about the false negative rate? Simulating reads is probably not necessary, but maybe the authors could analyze datasets which have been described in previous studies.

18) Since the authors identify filter criteria (line 276-279), it might be possible to calculate a score for predicted TE insertions. Some sort of ranking would be helpful to guide PCR validation approaches or the identification of cutoff values in specific applications.

19) Fig. 3: I am not convinced that the Pearson correlation coefficient is appropriate in these cases. Please check if the datasets follow a normal distribution and check the corresponding p-values of all Pearson correlation coefficients. Based on the data point distribution, I would assume that there is no correlation, but it is possible that datapoints are overlapping on the diagonal line. Also, the results of all 5 samples should be included as supplementary file.

20) The statement about sensitivity (line 297) is not supported by the presented results. The lowest analyzed coverage was >>60x. What would be the result for 30x or 20x re-sequencing data?

21) Please include a small dataset in the github repository. I would like to test TEfinder, but before running it on my own datasets, I would like to see that it works on a test dataset.

22) The authors thank MGHPCC for resources for a genome assembly process, but I have not read anything about this. Please check this and correct if necessary.

23) Please generate a DOI for the github repository content at zenodo: https://zenodo.org/.

24) How does the performance compare against other existing tools e.g. those described in the introduction. Usually, tools have different strengths and weaknesses. Therefore, it would be good to know how TEfinder compares to existing tools.

Author Response

REVIEWER 1

Sohrab et al. describe a shell script (TEfinder) for the detection of novel TE insertions based on a reference genome sequence and a paired-end read mapping. There are already several tools for the detection of TE insertions, but the novel objective of TEfinder is useability. I believe TEfinder could be a helpful solution for biologists who have only basic bioinformatics skills. However, there are still some issues and further demonstration of the capacities of TEfinder is needed.

 We would like to thank this reviewer for the comprehensive, insightful, and constructive criticisms and suggestions. Integration of many comments/suggestions helped to improve the usability of our tool and the clarity of this manuscript.

1) The analyzed fungal genome is small. The presented use case is also restricted to just one taxon. What is about larger data sets and different taxa e.g. plants and animals? I would like to see that TEfinder is also able to handle larger and different datasets. Therefore, I suggest to analyze these model systems in addition:

  1. a) The most important plant model species is Arabidopsis thaliana with Columbia-0 as the reference accession. Accession specific differences have been characterized e.g. for Niederzenz-1 (Nd-1) and there are even some PCR validations of such differences. It would be important to show that TEfinder is able to recover these. A TE annotation is available for Columbia-0 and should be a good test if TEfinder is able to handle any GFF file.

https://doi.org/10.1371/journal.pone.0164321

https://doi.org/10.1371/journal.pone.0216233

  1. b) The ultimate test is the analysis of an animal genome. I would suggest to take one of the latest phased long read assemblies and some randomly selected samples:

https://doi.org/10.1038/s41587-020-0719-5

https://doi.org/10.1073/pnas.1613365113

Alternatively the analysis of a Drosophila dataset could also show the applicability to animals:

https://doi.org/10.1093/nar/gkaa450

https://dx.doi.org/10.12688%2Ff1000research.9912.3

This is a good suggestion but requires substantial time. To be realistic, we decided to test TEfinder against one out of 220 Drosophila datasets from one suggested study (https://dx.doi.org/10.12688%2Ff1000research.9912.3). The good news is that we were able to run TEfinder through the data successfully and detected 63 insertions with default parameters. Unfortunately, the paper didn’t report TE instances. Therefore, we couldn’t find relevant information to draw conclusions on sensitivity and specificity and did not report this analysis in the manuscript.

In addition, we tested TEfinder on a subset of simulated Drosophila data from public database. The data included only one chromosome with 10 simulated insertions. The result is included in the manuscript.

“We used TEfinder with default options except ‘-k 30’ on the subset of simulated data from chromosome 2L in D. melanogaster. There were 10 simulated insertions and 8 were generated by TEs present in chromosome 2L. TEfinder identified all 8 insertion events. Since the other two simulated insertion events caused by TEs from other part of the genome and absent in the chromosome 2L TE annotation, this test did not detect them. Seven of the 8 events were in the “passed” category and 1 was labeled with strand biased supported by 32 forward and 14 reverse read counts (Supplementary File).”

2) When analyzing species where a long read genome assembly is available for multiple accessions, it is possible to perform a benchmarking through whole genome alignment. The authors might want to consider this as an alternative approach.

Another good suggestion. But our tool focuses on Illumina platform, as it is the cheapest sequencing option. Also, there are abundant Illumina data available in public domains.

3) Specific software versions are not necessarily an issue, but BioPython might not be able to process very large datasets if everything is loaded into memory. The authors might want to check if this would be another motivation for developing a new tool.

Thank you for recognizing this.

4) I do not understand the criticism about MELT. A FASTA and a BED file are common file formats and TEfinder also requires some kind of prior knowledge about TEs.

We have modified our manuscript to make it clear. The main issue with MELT is its complex input requirement. Users need to provide genome annotation and to create a set of files for each transposon presented as consensus sequence. MELT only accept “ATCG” in the consensus fasta file, other characters, such as, ‘N’, will confuse the alignment algorithms. For highly polymorphic TEs, users may have to manually inspect the consensus file. Users may be able to generate all required supporting files for each TE using RepBase or go through 6 steps which include running RepeatMasker, perform multiple sequence alignment, generating consensus sequence, and removing ambiguous sites. 

We changed the sentence to:

“The package is powerful in comprehensive TE analysis in human and chimpanzee genomes [12]. However, it requires gene annotations. In addition, for each TE, users need to provide consensus sequence of the TEs, and multiple files related to this particular transposon. Users have to make sure only “ATGC” characters are present in the consensus fasta file.”

5) The authors state that the main objective of TEfinder is useability. While this is a good justification for the development of a new tool, I still see some issues. BEDTools, SAMtools and Picard are already more requirements than e.g. Bowtie2.

  1. a) Is it possible to reduce the requirements to just samtools which is available as a jar file and does not require installation?

Another good suggestion! Unfortunately, the implementation will require much longer time than we have.

  1. b) The FASTA index file should be created by TEfinder to reduce the number of required input files.

Excellent idea! We have changed the input from FASTA index to FASTA.

  1. c) There should be instructions/suggestions for a very simple TE annotation approach. I assume this could be a major obstacle for the analysis of a novel genome assembly. Installing and running RepeatModeler or RepeatScout is probably not an option, because their use is challenging. Repbase is not freely accessible and a license is expensive. If users are able to deal with these tools, they could also use MELT for the following analysis. There needs to be a very convenient solutions which comes with an executable binary and does not require any installations.

This is a great point. Since this manuscript is not developed for TE annotation, we tried to improve our manuscript in following way:

The most straightforward tool is RepeatMasker [20], which is available on Galaxy and can be used to generate the GFF file based on the provided reference genome sequence and with a common repeat library from RepBase [17] or the TE library created above. The output from RepeatMasker should be filtered so that the simple repeats are removed.”

  1. d) TEfinder should support BAM and SAM files to make it easier to use. samtools could be used for internal conversion of SAM to BAM if necessary.

Agree! We have implemented an option for SAM input in addition to BAM.

  1. e) Mate pair reads (not just PE) should be supported. It is possible to convert them into paired-end. This increases the number of datasets that can be analyzed. Mate pair reads might even be beneficial when it comes to repetitive regions.

We agree that mate pair reads could be beneficial, especially when dealing with repetitive regions. However, such development will require the generation of suitable mate pair reads, in addition to code implementation. With the limited time granted to us, we decided to keep this version of TEfinder pair-end reads focused.

  1. f) TE names are probably a critical point, because spelling mistakes could cause TEfinder to break. Are there any forbidden characters which would break TEfinder like space, tab, comma, pipe, etc.? The authors might want to provide some additional scripts to generate this TE name list from the annotation file or do this internally to avoid another input. If a list is not specified, all TEs should be analyzed automatically.

Since each GFF/GTF file may have a different format in the attribute column, we decided not to offer the option to extract TE names from the file. Instead, we added more details on the manuscript.

“The TE names provided should exactly match the names in the GTF entries in a single column. The names cannot include characters other than letters, digits, ‘-’, ‘_’, and ‘#’.”

6) The authors state "TEs in GFF", but there are probably some more specific requirements. Was TEfinder tested with a diverse set of different GFF files? What is about GTF support? There are different versions of the GFF standard and their structure is surprisingly diverse.

We thank the reviewer for pointing this out. The TEfinder package supports both GFF2/GTF and GFF3 formats. Since the output of RepatMasker is GFF version 2, we decided to use GTF in the main text. We have added following statement in the manuscript:

“This file can be GFF2/GTF or GFF3”.

7) Is it possible to use multiple threads in TEfinder e.g. for samtools?

We added multiple threads for Samtools commands:

Moreover, multithreading is supported for the SAMtools commands of the pipeline via the thread option.”

8) The input is a GFF file, but the output is BED. It would be good to offer GFF as an alternative/additional output. Users might want to run down-stream analyses which require a GFF and not a BED file. The targeted users are probably not able to convert BED to GFF.

Agreed and we added GFF2/GTF output option.

9) There is a difference between the "reference genome" (DNA molecules) and the "reference genome sequence" (strings of characters, assembly). Since TEs are often not completely resolved in assemblies, it is crucial to differentiate between both. I noticed this issue throughout the manuscript. Please check the manuscript carefully and correct this.

We thank the reviewer for pointing this out and we corrected every instance of the ‘reference genome’ to ‘reference genome sequence’.

10) TEfinder uses paired-end information, but long reads (ONT, PacBio) should be even better. It would be possible to break long reads into paired-end fragments and feed them to TEfinder. The distances between both fragments would be precisely known. However, it might be better to have additional support for long reads. The authors might want to consider this, because most (re-)sequencing projects will use long reads in the future.

Although we agree with the reviewer, we decided to keep this version of TEfinder Illumina reads focused (See response in comment #5e).

11) Does the applied alignment tool influence the results? Some of the common aligners like Bowtie2, BWA-MEM, Novoalign, GEM3, etc. should be evaluated to ensure that TEfinder is working as expected. There might be differences concerning the alignment score, inclusion/exclusion of unpaired reads, and the handling of multi-mapped reads.

Although we didn’t have time to test different aligners as suggested here, we believe that as long as the “read group” is included in the bam file, TEfinder can perform equally well. All other aligners can include the read groups. Specifically, TEfinder ignores unpaired reads. Alignments scores are also ignored. If multi-mapped reads are not primary reads, they are filtered out. Otherwise, all of them are used.

We added detail about read groups in the text:

“Users may choose other aligners, as long as read group information is added in the BAM header, created by each aligner. The aligned BAM files need to be sorted by coordinates. The bwa command above specifies read group information using bwa-mem, option -R. If Bowtie2 is used, users need to select option --rg-id. For GEM3-Mapper, option -r need to be specified. If read group information was not included in the initial alignments, Picard has AddOrReplaceReadGroups function to create read group information based on the input BAM file.”

12) What is about reads mapping in opposite directions or on different contigs? The pipeline description does not mention these cases.

Both reads mapped to the opposite directions and different contigs are considered as discordant reads and used to find TE insertion instances. Specifically, “the corresponding mate maps to a different sequence or the read pair has an insert size that exceeds a threshold of 10 times the mean insert size”. Different contigs and opposite directions (if the absolute insert size was higher than the mean insert size) are included.

13) Is there a minimal coverage that is required for this analysis? Is it possible to set a minimal coverage as filter? It would be good to evaluate subsets of reads to identify a minimal required coverage for accurate results.

Excellent point! To take this into considerations, we conducted additional analysis and included an additional short section “3.3.Sensitivity and Applicability of TEfinder”. 

 As expected, the lower the sequence coverage, the less sensitive it is in identifying TIPs. However, at 20X sequence coverage, TEfinder is still able to detect more than 50% TIP events comparoing to ~100x sequence coverage used in our initial data analysis. Therefore, we suggested a minimal sequence coverage to be =>20X for a mixed population.

To ensure effective in detect TIPs, we suggest a minimal 20x sequence coverage, if the reads are generated from a mixed population.”

14) Are nested TEs an issue for TEfinder? The two groups of reads might not be clear if there are multiple TE copies inserted nested into an existing TE array. I think this is partly addressed by the two output files, but the authors might want to provide some additional details.

Thank you for point this out. To increase clarity, we added following statement in the discussion section:

“TEfinder depends on read placement within a genome. Ambiguity exists in such processes especially when placing a read into a highly repetitive region, which may result in lower confidence level in calling a TIP.  Such insertion events, as well as nested TEs, might still be detected and reported by TEfinder with a tag ‘in repeat’, to differentiate them from TIP events with high confidence.”

15) The authors might want to add some additional details about the core concept of identifying and filtering novel TE insertions. Are there any cutoff values involved?

We thank the review for this suggestion. We combined two BED outputs into one and added one more tag in the 7th column of the output: “FILTER”. The events previously reported in ‘TEinsertions_inRepeatRegions.bed’ file, now have the “in_repeat” filter. Furthermore, we added “weak_evidence” for the events with read counts less than 10. Finally, we introduced “strand_bias” filter for the events that have high forward/reverse read count disparities. We updated the main text and Figure 1A accordingly.

16) The run time looks reasonable. The details about memory usage are fine, but how many cores were used? Is it possible to run analyses in parallel e.g. run analysis for different contigs in parallel?

We added the core information in the text:

On a high-performance computing cluster with a memory request of 3 cores X 50000 MB, the trial runs took an average of 4.7 hours to complete with the minimum time being 3.6 h across the samples.”

17) How many of the inspected insertions were correctly predicted by TEfinder? This would reveal the true positive rate. What is about the false negative rate? Simulating reads is probably not necessary, but maybe the authors could analyze datasets which have been described in previous studies.

To increase clarity of our manuscript, we have report comprehensive PCR validation of TIP instances applying both filters (requiring minimal 50 mapped reads and only report TIPs in well assembled unique genomic regions) and added a short paragraph below.

“We selected 13 newly detected TIPs in the “PASS” catalog for experimental validation (Table 2). Results of PCR confirmed our prediction in 12 out of 13 cases, resulting in a successful rate of 92% (the failed one was a low frequency event with 22% allele frequency in the population)”

We also used simulation data, ee response in comment #1.

18) Since the authors identify filter criteria (line 276-279), it might be possible to calculate a score for predicted TE insertions. Some sort of ranking would be helpful to guide PCR validation approaches or the identification of cutoff values in specific applications.

We thank the reviewer for this suggestion. As we mentioned above, we added filter tag (column 7) to the output file. In addition, the 5th column of the BED output, where quality scores usually recorded, reports the read count. This reporting system enables users to make decision based on either read count and/or the internal filtering.

We changed the main text and the Table 2 accordingly:

 “The identified insertion sites are filtered based upon 3 criteria: occurrence in repeat regions, supporting read count, and strandness. If the insertion site coincides with a TE annotated site in the reference genome sequence, its filter label becomes ‘in_repeat’. If the total supporting reads for the insertion event is less than the cutoff 10, then the filtering process will label this low confidence event as ‘weak_evidence’. Strand bias is critical for filtering out insertion events. Two threshold values are calculated based on the forward read count (F), and if the reverse read count (R) falls outside those boundaries (R < F0.8 or R > F1.25), then the event likely is not caused by TIP and gets ‘stand_bias’ label. If an insertion site passes all three filtering criteria, then it will be labeled ‘PASS’, representing high confidence insertion events (Figure 1).”

19) Fig. 3: I am not convinced that the Pearson correlation coefficient is appropriate in these cases. Please check if the datasets follow a normal distribution and check the corresponding p-values of all Pearson correlation coefficients. Based on the data point distribution, I would assume that there is no correlation, but it is possible that datapoints are overlapping on the diagonal line. Also, the results of all 5 samples should be included as supplementary file.

After considering this comment carefully, we removed Figure 3 and related text. Instead, we included discussion on the new internal filtering and how it uses evidence read counts, and the forward/reverse read count bias to guide the users. We included all output bed files generated in this study in a zip file as supplementary data.

20) The statement about sensitivity (line 297) is not supported by the presented results. The lowest analyzed coverage was >>60x. What would be the result for 30x or 20x re-sequencing data?

Excellent suggestion. We have included a short section “3.3. Sensitivity and Applicability of TEfinder”.  As expected, the lower the sequence coverage, the less sensitive it is in identifying TIPs. When we reduced the read coverage to half at ~50X sequence coverage, we are able to capture 82% of the events (and 82% of the events which passed internal filtering). At 20X sequence coverage, TEfinder is still able to detect more than 50% TIP events comparing to ~100x sequence coverage used in our initial data analysis. In the dataset with ~10x coverage, the total detected events reduced to 30% (38% passed internal filtering). To make it clear, we included “The sensitivity of TEfinder depends on the read coverage of the alignments in the input BAM file” and suggested a minimal 20X sequence coverage.

21) Please include a small dataset in the github repository. I would like to test TEfinder, but before running it on my own datasets, I would like to see that it works on a test dataset.

That is fantastic! We provided one of our test samples in the GitHub repository with other required inputs: https://github.com/VistaSohrab/TEfinder/tree/master/test_dataset

22) The authors thank MGHPCC for resources for a genome assembly process, but I have not read anything about this. Please check this and correct if necessary.

Thanks for catching this up. We changed the acknowledgement to “We thank the Massachusetts Green High Performance Computing Center for providing high-performance computing capacity for the development and testing of our tool.

23) Please generate a DOI for the github repository content at zenodo: https://zenodo.org/.

Thank you for the suggestion and we generated DOI for the github repository: http://doi.org/10.5281/zenodo.4446971.

24) How does the performance compare against other existing tools e.g. those described in the introduction. Usually, tools have different strengths and weaknesses. Therefore, it would be good to know how TEfinder compares to existing tools.

A good idea. But it is not possible for us to implement within the timeframe we have.

Reviewer 2 Report

In the article authors have proposed a pipeline, TEfinder, for detecting new transposable element insertion events in NGS data. As per authors, the main advantage of their pipeline is less dependency and expertise requirement. It is a UNIX command line pipeline which required other tools, including but not limited to Samtools, Bedtools and Picard.

In my opinion, the work has not much novelty as it is just combination of a few tools. Although TFfinder could be useful for a few specific cases, I do not find any solid reason due to which someone would like to choose TFfinder over their current tool/pipeline.

Following are may major comments and suggestions:

  1. Although there are more than 10 tools out there, authors have picked only two tools for TEfinder’s advantages. Moreover, those advantages are not appealing. .eg. python (an open source softare) requirement for ISmapper should not be an issue for researcher who could work in UNIX command line. Similarly, MELT requires java (same as picard in TFfinader) and cannot be used for unmodeled organism which lack well-annotated genomes. TEfinder also works fine for well annotated organism, but for others, it requires another tool (RepeatModeler or RepeatScout) and reference genome (thus more dependencies). The pipeline could have other advantages, e.g., low RAM requirement, less computation time, better sensitivity etc, but the authors have not included them.
  2. It would be better if authors compare current pipeline results with other available tools in term of computational efficiency, sensitivity, specificity etc.
  3. Although authors have validated four detected TE insertion events by PCR, it is less than 1% of 500 plus total detected   It would be better if authors report statistical test results showing accuracy of the pipeline based on number of true and false positives.

Author Response

REVIWER2

In the article authors have proposed a pipeline, TEfinder, for detecting new transposable element insertion events in NGS data. As per authors, the main advantage of their pipeline is less dependency and expertise requirement. It is a UNIX command line pipeline which required other tools, including but not limited to Samtools, Bedtools and Picard.

In my opinion, the work has not much novelty as it is just combination of a few tools. Although TFfinder could be useful for a few specific cases, I do not find any solid reason due to which someone would like to choose TFfinder over their current tool/pipeline.

We would like to thank this reviewer for your time.

TEfinder was developed due to various challenges we encountered in applying existing tools. For this revise manuscript, we also tested TEfinder against publicly available Drosophila data successfully. We included the testing result of TEfinder on a set of Drosophila simulation data, which revealed high sensitivity and specificity of TEfinder. We are highly confident that this tool can be useful for many systems other than fungi.

Following are may major comments and suggestions:

  1. Although there are more than 10 tools out there, authors have picked only two tools for TEfinder’s advantages. Moreover, those advantages are not appealing. .eg. python (an open source softare) requirement for ISmapper should not be an issue for researcher who could work in UNIX command line. Similarly, MELT requires java (same as picard in TFfinader) and cannot be used for unmodeled organism which lack well-annotated genomes. TEfinder also works fine for well annotated organism, but for others, it requires another tool (RepeatModeler or RepeatScout) and reference genome (thus more dependencies). The pipeline could have other advantages, e.g., low RAM requirement, less computation time, better sensitivity etc, but the authors have not included them.

Thanks for pointing these out. We have improved the manuscript by increasing clarity and specifying additional advantages that TEfinder offers.

  1. It would be better if authors compare current pipeline results with other available tools in term of computational efficiency, sensitivity, specificity etc.

It could be an interesting study as suggested. But a carefully designed comparative study will require substantial efforts that is beyond the scope of this manuscript.

  1. Although authors have validated four detected TE insertion events by PCR, it is less than 1% of 500 plus total detected   It would be better if authors report statistical test results showing accuracy of the pipeline based on number of true and false positives.

Thanks for pointing this out. We changed how we report the TIP events in the output.

The number 500 was derived from the raw outputs, which are further filtered upon 3 criteria: occurrence in repeat regions, low supporting read count, and strand bias. To increase clarity of our manuscript, we have added filter tag (column 7) to the output file, in addition to the 5th column of the BED output (read count), where quality scores usually recorded. This reporting system enables users to make decision based on either read count, or uniqueness of the region or combining both. We applied filters that first removed the ones present in the WT, and then the ones in repetitive regions. The high confidence insertion events are labeled in the ‘PASS’ category and we only PCR validated those with high confidence. We included following:

“We selected 13 newly detected TIPs in the “PASS” catalog for experimental validation (Table 2). Results of PCR confirmed our prediction in 12 out of 13 cases, resulting in a successful rate of 92%. The one TIP event that failed PCR amplification only has 22% allele frequency in that evolved population.”

Round 2

Reviewer 1 Report

The authors did a great job in adding new functions and generally improved TEfinder. A very user friendly tool for a convenient TE analysis would be important for the community. However, the advantages over existing tools and applicability to large datasets need more convincing demonstration. I would suggest the authors just take the time they need to perform these additional analyses. I understand that it might not be possible to do this within 10 days. Also, I was not able to run TEfinder successfully. This might be related to the compression of the BAM file in the example dataset, but the authors should check if they can find an issue.

1) It is good to see that the analysis of one Drosophila sample was possible. However, users are probably interested to find the differences between all 220 samples. If it is not possible to present analyses of model organisms, because the run time does not permit it, I would suggest the authors improve the performance of TEfinder first. Users would expect that it is possible to run an analysis on their sample within hours or maybe a few days. The A. thaliana genome has an extremely small size of <150Mbp. Most crop plants have 10x larger genomes. Obviously, the authors should look for other datasets which would allow some kind of external validation. Genes is a broad journal and readers might be interested to see a demonstration on a species "close" to their own species of interest.

2) This suggestion was intended to perform an easy benchmarking. I would like to add that most sequencing projects in the future are likely to use (only) long reads. It is also no longer correct, that Illumina is generally cheaper. Therefore, I would like to encourage the authors to reconsider the integration of support for long reads.

3) ok

4) ok

5) a) Please take the time to make TEfinder as easy to use as possible. This would substantially improve the tool, because useability is the major novelty.

b) ok

c) This is still complicated and some users might not be allowed to upload their data. I think a really convenient tool (or event a functionality of TEfinder) is required.

d) ok

e) ok

f) ok, but a functionality in TEfinder to clean the GFF file would be even better.

6) ok

7) ok

8) ok

9) ok

10) ok

11) The explanations are very helpful. I think an evaluation of different aligners would still be important. There might be performance differences with respect to the TE identification, because there are also differences with respect to other applications like variant detection.

12) ok

13) ok, but I would suggest to include a figure where the percentage of detected events is plotted against difference coverage values.

14) ok

15) ok

16) The additional information is helpful, but information about the option to process individual contigs in parallel would be great. The authors might want to consider offering a functionality to just analyze a single contig/chromosome.

17) ok. The authors might want to consider adding a statement that the success rate might be different (lower?) in repetitive regions though.

18) ok

19) ok

20) ok

21)

a) The authors might want to add support for gzip compressed FASTA and GFF files.

b) An error occurred when running TEfinder on the test dataset:

[E::hts_open_format] Failed to open file xxx/Y2.fixmate.dedup.bam.gz

samtools sort: can't open "xxx/Y2.fixmate.dedup.bam.gz": Exec format error

The authors might want to ask colleagues to test TEfinder. This could also reveal issues in the documentation.

22) ok

23) ok

24) This is very important for potential users. Even with useability being the main novelty of TEfinder, it would be important to show that it performs similar to established tools.

Author Response

RESPONSE TO REVIEWER 1

The authors did a great job in adding new functions and generally improved TEfinder. A very user friendly tool for a convenient TE analysis would be important for the community. However, the advantages over existing tools and applicability to large datasets need more convincing demonstration. I would suggest the authors just take the time they need to perform these additional analyses. I understand that it might not be possible to do this within 10 days. Also, I was not able to run TEfinder successfully. This might be related to the compression of the BAM file in the example dataset, but the authors should check if they can find an issue.

We appreciate the recognition of the importance of our tool by this reviewer and for the efforts to test it. We would like to apologize for the error this reviewer encountered while running our testing dataset. The problem was caused by a corrupted file introduced during the data uploading process. As we were in a rush, we didn’t have the time to run the test. The problem was easily fixed by replacing all testing files in GitHub. Several lab mates have successfully tested the program using newly provided testing files.

We would like to thank this reviewer for additional comments to improve the manuscript and the TEfinder program. Please see specific responses below.

It is good to see that the analysis of one Drosophila sample was possible. However, users are probably interested to find the differences between all 220 samples. If it is not possible to present analyses of model organisms, because the run time does not permit it, I would suggest the authors improve the performance of TEfinder first. Users would expect that it is possible to run an analysis on their sample within hours or maybe a few days. The A. thaliana genome has an extremely small size of <150Mbp. Most crop plants have 10x larger genomes. Obviously, the authors should look for other datasets which would allow some kind of external validation. Genes is a broad journal and readers might be interested to see a demonstration on a species "close" to their own species of interest.

We didn’t analyze 220 Drosophila samples, simply because the original data did not report TIPs for a meaningful comparison. However, we took the advice from this reviewer and found an Arabidopsis study (The Arabidopsis thaliana mobilome and its impact at the species level DOI: 10.7554/eLife.15716), where authors identified non-reference TE insertions in several Arabidopsis accessions. We selected 2 accessions with TE insertion events in the middle range of all samples they tested. We downloaded their data, mapped the raw reads against the reference genome, and successfully detected TIPs using TEfinder. We summarized our findings in the Results section as following:

Furthermore, we tested TEfinder with data used in the Arabidopsis thaliana mobilome study [29] that detected 63 non-reference TE insertion events in Alst-1 and 61 in Benk-1 accessions, respectively. With default options, TEfinder successfully detected 55, and 58 of these known instances, respectively. These testing results demonstrated feasibility in applying TEfinder in plants, animals (refer to Drosophila data we tested in the 1st revision), and fungi with high sensitivity.”

We manually inspected 8 missing TEs in Alst-1 dataset: two of them didn’t have any supporting read evidence, one of them had only forward read evidence, 4 events were small insertions with less than 20 bases instead of TE insertion, and one of the TEs wasn’t annotated in the input GTF file.

This suggestion was intended to perform an easy benchmarking. I would like to add that most sequencing projects in the future are likely to use (only) long reads. It is also no longer correct, that Illumina is generally cheaper. Therefore, I would like to encourage the authors to reconsider the integration of support for long reads.

The implementation of this new function will require new logic in the design.

This is still complicated and some users might not be allowed to upload their data. I think a really convenient tool (or event a functionality of TEfinder) is required.

Galaxy is web-based and specifically aimed at not advanced users.

a functionality in TEfinder to clean the GFF file would be even better.

We included example command lines to clean the RepeatMasker output and to generate the input text file from the GTF file in the tool manual.

The explanations are very helpful. I think an evaluation of different aligners would still be important. There might be performance differences with respect to the TE identification, because there are also differences with respect to other applications like variant detection.

We mapped one of the fungal datasets with Bowtie2 aligner using default options. Although the number of detected raw instances were doubled (Bowtie2: 1107, BWA: 567), the PASS events were almost identical. We added the following statement in the manuscript:

“Different aligners can have slightly different results.”

ok, but I would suggest to include a figure where the percentage of detected events is plotted against difference coverage values.

We added new Figure 3 to the manuscript.

“Percent of detected TIP events with respect to read coverage in the input alignment file. Orange circles depict all reported events while blue triangles represent PASS events.”

The additional information is helpful, but information about the option to process individual contigs in parallel would be great. The authors might want to consider offering a functionality to just analyze a single contig/chromosome.

Because of our TIP search algorithm, adding such a feature would increase the run time. Therefore, we decided not to implement this option.

  1. The authors might want to consider adding a statement that the success rate might be different (lower?) in repetitive regions though.

Thanks, and we included the statement in the manuscript.

Please take the time to make TEfinder as easy to use as possible. This would substantially improve the tool, because useability is the major novelty.

We included the gunzip command in the manual. We fixed the example input files in GitHub and confirmed that they are not corrupted.

This is very important for potential users. Even with useability being the main novelty of TEfinder, it would be important to show that it performs similar to established tools.

We had tried running ISmapper, MELT, TEMP but we encountered problems in installation or running, further emphasizing the usability of our program.

Reviewer 2 Report

Respone:  “TEfinder was developed due to various challenges we encountered in applying existing tools. For this revise manuscript, we also tested TEfinder against publicly available Drosophila data successfully. We included the testing result of TEfinder on a set of Drosophila simulation data, which revealed high sensitivity and specificity of TEfinder. We are highly confident that this tool can be useful for many systems other than fungi.”

Authors have added Dorsophila results which has expanded it capability. However, the advantages of TEfinder still do not stand out.

Response: “We have improved the manuscript by increasing clarity and specifying additional advantages that TEfinder offers”

Although authors have added/modified many lines and numbers, all the edits in ‘track changes’ doesn’t add to clarity.  Further, the only new advantage that I noticed is in Line 409: The pipeline has low RAM requirement and takes rather short computation time.” Also “with a memory request of 3 cores X 50000 MB, the trial runs took an average of 4.75.2 hours”

50gb doesn’t seems low memory, especially for such small chromosomes. Can authors provide ‘computational requirement’ of other tools for same input? Or, how much time/ram would TEfinder require for dataset used in “https://doi.org/10.1186/s13100-019-0197-9”

Response: “It could be an interesting study as suggested. But a carefully designed comparative study will require substantial efforts that is beyond the scope of this manuscript.”

In my opinion, comparison with 1 or 2 tools should not take much resource/time. But it would considerably increase the confidence in the tool

Response: “We applied filters that first removed the ones present in the WT, and then the ones in repetitive regions. The high confidence insertion events are labeled in the ‘PASS’ category and we only PCR validated those with high confidence.”

In the new results, out of total 2747 events in 5 samples 302 were marked pass.  Further, 13 of them were tested for experimental validation.

What’s the overlap (or reproducibility) of earlier ~500 events with new ‘Pass events’?

On what criteria the PASS event selected for validation?

Why 1/63 selected from Y2 and 7/64 from Y5?

Unless there is a specific criterion behind selection, it would be better to randomly select about 20% (10) from each sample.

Author Response

RESPONSE TO REVIEWER 2

We would like to thank this reviewer for their time and comments.

Authors have added Dorsophila results which has expanded it capability. However, the advantages of TEfinder still do not stand out.

In addition to the Drosophila dataset, we successfully ran TEfinder on Arabidopsis thaliana datasets as reported in section 3.3. of our results.

“Furthermore, we tested TEfinder with data used in the Arabidopsis thaliana mobilome study [29] that detected 63 non-reference TE insertion events in Alst-1 and 61 in Benk-1 accessions. With default options, TEfinder successfully detected 55, and 58 of these known instances, respectively. These testing results demonstrated feasibility in applying TEfinder in plants, animals, and fungi with high sensitivity.”

The main advantages of TEfinder are fewer input file requirements and software dependencies relative to other tools.

Although authors have added/modified many lines and numbers, all the edits in ‘track changes’ doesn’t add to clarity.  Further, the only new advantage that I noticed is in Line 409: The pipeline has low RAM requirement and takes rather short computation time.” Also “with a memory request of 3 cores X 50000 MB, the trial runs took an average of 4.75.2 hours”

50gb doesn’t seems low memory, especially for such small chromosomes. Can authors provide ‘computational requirement’ of other tools for same input? Or, how much time/ram would TEfinder require for dataset used in https://doi.org/10.1186/s13100-019-0197-9

This reviewer should find a pdf file with all track changes accepted. Submitting a document with all changes tracked is for all reviewers to easily identify what modifications we have implemented by considering all comments/suggestions.

This revision tested the Arabidopsis genome, which is not so small.

In my opinion, comparison with 1 or 2 tools should not take much resource/time. But it would considerably increase the confidence in the tool

We wish all tools are easy to use, unfortunately, this is not the case.

In the new results, out of total 2747 events in 5 samples 302 were marked pass.  Further, 13 of them were tested for experimental validation.

What’s the overlap (or reproducibility) of earlier ~500 events with new ‘Pass events’?

The results in the first version of the manuscript and the revised version are the same except for how we report the events. Initially, we reported our results in two files, in the revised version, we combined these two files and included additional filtering information.

The word “new” in line 290 probably caused this confusion, and we have deleted that word. What this sentence referred to is the insertion events detected in the evolved population. TEfinder reports raw output, which is much larger because this output includes many instances in the repetitive regions. The Pass events are instances after applying several filters. As stated in the text between lines 290-296:

As expected, many insertion events are detected among complex repetitive regions that capture most transposons. In fact, the number of TE insertions reported in each sample population is approximately one third of the insertions detected in known repeat regions. Additionally, there are filtered out events because of the low read count (‘weak evidence’). Furthermore, almost half of the detected events have strand bias as estimated by a power function.”

On what criteria the PASS event selected for validation? Why 1/63 selected from Y2 and 7/64 from Y5? Unless there is a specific criterion behind selection, it would be better to randomly select about 20% (10) from each sample.

Yes, there is a very strong justification, Y5 has the most intriguing biological phenotypes compared to all other evolving strains.
